# Biological Applications of Silica-Based Nanoparticles

**Franciele da Silva Bruckmann** [1], **Franciane Batista Nunes** [1], **Theodoro da Rosa Salles** [1], **Camila Franco** [2], **Francine Carla Cadoná** [2] and **Cristiano Rodrigo Bohn Rhoden** [1,3,*]

1   Laboratório de Materiais Magnéticos—LaMMaN, Universidade Franciscana—UFN, Santa Maria 97010-030, RS, Brazil
2   Mestrado em Ciências da Saúde e da Vida, Universidade Franciscana—UFN, Santa Maria 97010-030, RS, Brazil
3   Programa de Pós-Graduação em Nanociências, Universidade Franciscana—UFN, Santa Maria 97010-030, RS, Brazil
*   Correspondence: cristianorbr@gmail.com

**Abstract:** Silica nanoparticles have been widely explored in biomedical applications, mainly related to drug delivery and cancer treatment. These nanoparticles have excellent properties, high biocompatibility, chemical and thermal stability, and ease of functionalization. Moreover, silica is used to coat magnetic nanoparticles protecting against acid leaching and aggregation as well as increasing cytocompatibility. This review reports the recent advances of silica-based magnetic nanoparticles focusing on drug delivery, drug target systems, and their use in magnetohyperthermia and magnetic resonance imaging. Notwithstanding, the application in other biomedical fields is also reported and discussed. Finally, this work provides an overview of the challenges and perspectives related to the use of silica-based magnetic nanoparticles in the biomedical field.

**Keywords:** magnetic nanoparticles; multifunctional systems; nanocarriers





## 1. Introduction

Nanoscience and nanotechnology explore the properties and application of particulate systems at the nanometric size. At this scale, materials exhibit properties and characteristics different from their bulk form [1] due to the surface and quantum confinement effects [2]. These effects are related to the increase in the area/volume ratio, which can improve textural properties, such as specific surface area and porosity. Additionally, quantum effects are involved with electronic and optical modifications [3–5]. When considering their unique properties and characteristics, nanoparticles and nanomaterials have been widely used in water remediation [5], pesticide detection [6], and especially in biomedical applications [7]. Despite their excellent properties, the toxicity of nanoparticles (NPs) is a common concern for the scientific community [8].

Among the nanoparticulate systems, silica nanoparticles (SiNPs) stand out for their excellent properties for biological applications. Inorganic SiNPs have a high surface area, biocompatibility, and stability in tissues under acid conditions, improving the thermal and physicochemical properties and preventing other compounds from erosion and thermal degradation [9,10].

In addition, silica is approved by the Food and Drug Administration (FDA) due to its inertness, low toxicity, thermal resistant characteristic, and biocompatibility. The presence of silanol groups (Si-OH) in its structure allows easy complexation with other materials by covalent bondings, such as antibody attachment, nucleic acids, and fluorescent molecules [10–14]. According to the structural form, nanosilica can be classified as amorphous silica, mesoporous silica (pore size ranging from 2 to 50 nm), and silica-gels, among others [15]. In biological systems, pore diameter, particle size, shape, and chemical modification can affect biodistribution and toxicity [16]. Although silica nanoparticles exhibit excellent properties, some studies reported that silica nanoparticles exhibit significant

challenges that should be given importance. In this scenario, some investigations suggested that this material can show toxicity and adverse effect depending on the type of cell and concentration [17,18].

For instance, this material can be distributed throughout the circulatory system, reaching different tissues of the human body and causing systemic damage. A recent review reports the potential toxicity to the respiratory, nervous, and digestive systems. The mechanisms of toxicity depend on many factors, but in most cases, the damage is associated with oxidative stress, inflammation, and apoptosis [9]. Liu et al. [19] reported that silica nanoparticles exhibit cytotoxicity in endothelial cells via oxidative stress production by JNK/p53 and NF-kB pathways triggering and possibly increasing the proinflammatory and procoagulant responses via CD40-CD40L-mediated monocyte–endothelial cell interactions activation.

Moreover, another investigation indicated that silica nanoparticles could cause MAPK/Nrf2 pathway and nuclear factor-κB signal transduction in endothelial cells, producing oxidative stress and inflammation and affecting the nitric oxide (NO)/nitric oxide synthase (NOS) system [20]. Moreover, Petrick et al. [21] found that silica nanoparticles generated a pro-atherogenic effect in mouse cell line macrophage J774 by increasing oxidative stress, cytotoxicity, and triglyceride. In agreement with these studies, Guo et al. [22] reported that silica nanoparticles could induce lipids accumulation in macrophages, stimulate oxLDL-induced foam cell development, and trigger macrophage apoptosis.

In addition, an important review conducted by Song et al. [23] reported that the main challenge of silica nanoparticles focuses is drug delivery. This study indicated that more research should be performed regarding thermos-responsive nanocarriers that present safe and sensible profiles to respond to slight temperature variations nearby the biological temperature of 37 °C.

Silica is widely used for coating magnetic nanoparticles (MNPs), such as magnetite and ferrite-based transition metals, due to its excellent properties allowing the reduction in the limitations generally presented by MNPs, such as potential toxicity and aggregation, resulting from the high total surface energy [24]. The coating not only improves biocompatibility and physicochemical stability but also enables the synthesis of multifunctional particles [12,25].

Despite the drawbacks, MNPs show excellent responsiveness to an external magnetic field, high relaxivity, and surface area, permitting their use in biomedical applications [26]. Previous investigations reported that silica-based magnetic nanoparticles present several biological and therapeutical proprieties, such as antitumor, anti-protozoal, antibacterial, and stroke treatment [27–33]. These biomedical activities can be explained by their remarkable capacity to improve drug release and therapeutic procedures [34,35]. Moreover, some studies indicated that silica nanoparticles exhibit excellent biocompatibility and biosafety profile since they present low or absence of side effects [36,37]. Another important activity is the application of silica nanoparticles to improve imaging assessment and enhance the diagnosis in nanomedicine [37,38]. The main biological properties of silica nanoparticles are summarized in Figure 1.

This review reports the main aspects related to the biological applications of silica-based magnetic nanoparticles. The safety profile and biocompatibility using in vivo and in vitro models are also presented, furnishing insights regarding the influence of chemical modifications on toxicity. Moreover, recent advances in drug delivery systems, pH and thermal-responsiveness, combined therapy, and nanotheranostic are highlighted. The enhancement in antioxidant and antimicrobial action, as well as the drug repositioning, are mentioned and discussed.

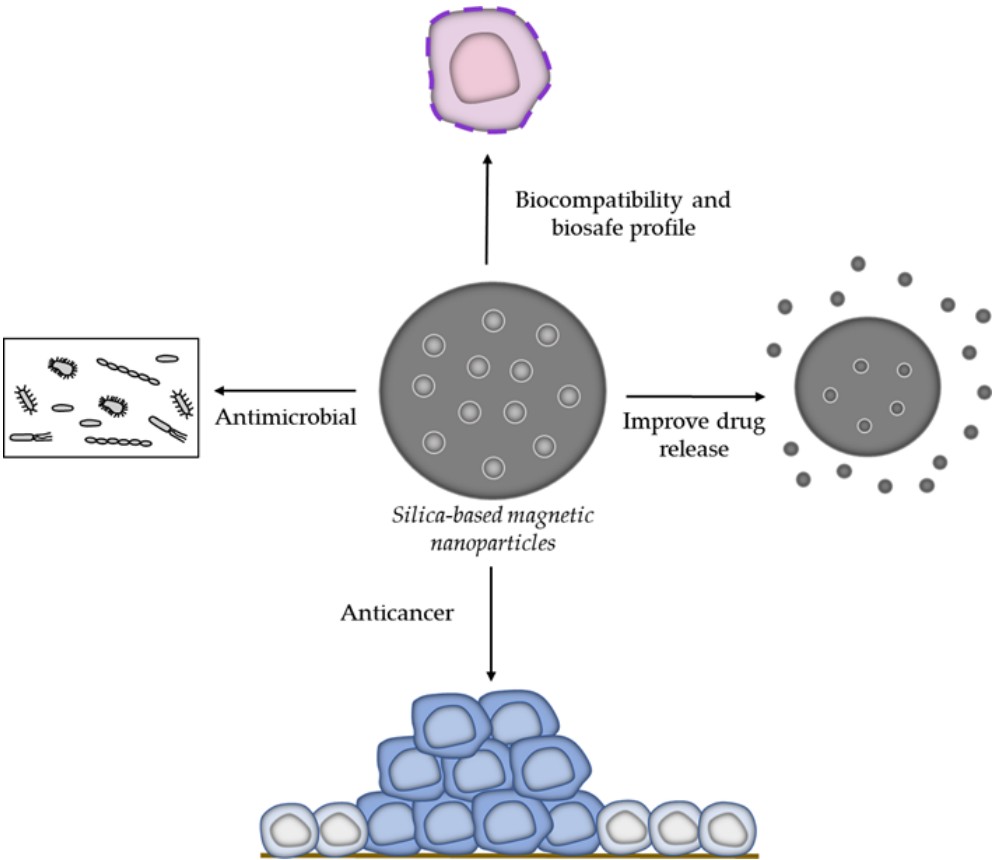

**Figure 1.** Biological properties of silica nanoparticles.

## 2. A Brief Approach to the Methods of Obtaining Silica-Based Nanoparticles

The strategies used in the synthesis of silica-based magnetic nanoparticles are related to their intrinsic characteristics, such as size, shape, and textural and magnetic properties. This section reports a brief review of the main methods used to obtain silica-based magnetic nanoparticles, as well as their characteristics and properties. In this scenario, several approaches were employed, such as sol–gel [39,40], chemical modification (surface functionalization) [41], inverse microemulsion [42], reverse microemulsion [43], impregnation [44], hydrolysis, and condensation [45]. Furthermore, different chemical routes were explored to produce magnetic nanoparticles incorporated, modified, and decorated with silica nanoparticles. Methods such as co-precipitation [46], thermal decomposition [45], and sol–gel are used to synthesize ferrites and silica-based magnetic nanoparticles (copper, manganese, zinc, nickel, and cobalt).

The main approach for obtaining iron oxide nanoparticles includes co-precipitation. A straightforward and simple method that involves the precipitation of ferrous and ferric ions under an inert atmosphere, controlled pH (pH 9–14), and mild conditions [24,35,47–49]. An innovative modification of this method was recently reported by Rhoden et al. [50]. This report focused on the use of only one iron source ($Fe^{2+}$) without the need for complex reaction systems (controlled medium). In addition, the novel approach allows controlling the amount of incorporated iron oxide nanoparticles into the surface of other materials.

The calcination technique is an important and widely explored tool for synthesizing silica-based nanoparticles. Khanna et al. [10] investigated the effect of calcination temperature on the crystallinity, size, and morphology of copper ferrite nanoparticles. The results found showed a correlation between the temperature and particle size. Moreover, non-calcined samples exhibit non-defined morphology and amorphous nature. El-Sayyad et al. [46] developed multicore–shell nanoparticles at high calcination tem-

peratures (450 °C). The nanocomposite presented a semi-spherical shape with an average diameter ≈ of 125 nm.

Similar behavior was observed by Keshavarz et al. [35] when synthesizing magnetic nanoparticles with a thin layer of oleic acid coated with mesoporous silica (OA-$Fe_3O_4$@$mSiO_2$) through the methodology employing tetraethyl orthosilicate (TEOS) and surfactant (cetyltrimethylammonium-bromide—CTAB). The poly (carboxy betaine methacrylate) (CBMA)-functionalized nanoparticles showed a larger particle size (increased from 140 to 200 nm). In addition, OA-$Fe_3O_4$@$mSiO_2$ and OA-$Fe_3O_4$@$mSiO_2$ modified with pCBMA (OA-$Fe_3O_4$@$mSiO_2$@pCBMA) exhibited lower saturation magnetization values compared to pure magnetite. Thermo-responsive nanoparticles were synthesized through in situ hybridization. The characterization results show that nanoparticles exhibited a spherical morphology with a slight decrease in crystallinity and saturation magnetization value due to polymeric and silica coating (the $M_s$ values decreased around 28 emu g$^{-1}$ with the coating) [49].

Oleic acid-stabilized $Fe_3O_4$ nanoparticles were synthesized through thermal decomposition, while the silica-coating was carried out by reverse microemulsion. The nanoparticles showed a high value of saturation magnetization and coercivity equal to zero ($H_C = 0$), indicating that ($Fe_3O_4$@$SiO_2$) presents a superparamagnetic behavior. In addition, when coated with silica employing the microemulsion technique furnishes hydrophilic characteristics to iron oxide nanoparticles, the interaction with the biological environment improved [51]. Superparamagnetic behavior was verified in mesoporous silica-coated cobalt ferrites [52]. Nanoparticles presented different magnetic properties (saturation magnetization, coercivity, and remnant magnetization) considering the calcination technique (conventional oven or microwave). The calcination type also affects the morphology, size distribution, and crystallite size.

Hydrophobic $Fe_3O_4$ nanoparticles were synthesized by the hydrothermal method, followed by coating with fluorescent silica using the reverse microemulsion technique [37]. Through transmission electron spectroscopy images, the nanoparticles had a diameter of around 25 nm, with the magnetic core in a spherical shape, while hysteresis loops display that magnetic nanoparticles exhibit superparamagnetic behavior. The hydrothermal process was also employed to produce monodisperse magnetite nanoparticles [53].

In order to avoid its agglomeration and oxidation, the Stöber method (hydrolysis of TEOS) was used to coat the magnetic core, reducing the physicochemical instabilities. Scanning electron microscopy images showed that $Fe_3O_4$@$SiO_2$ nanoparticles demonstrated a spherical characteristic and average size of around 340 nm. X-ray diffraction patterns displayed an amorphous band at $2\theta \approx 22°$, which refers to silica-coating.

Recently, silica-based double-layer core–shell structures were developed by combining hydrothermal and Stöber synthesis [54]. Additionally, surface modification with silver was performed by the alkaline etching process. XRD patterns revealed that the coating and etching procedures did not cause changes in crystalline phases. At the same time, the TEM images showed spherical and monodisperse particles, and core–shell structure could be observed.

Sol–gel and green synthesis were used to produce silica-coated magnetic nanoparticles ($Fe_3O_4$@$SiO_2$) and $Fe_3O_4$@$SiO_2$ decorated with silver and gold, respectively. The hybrid systems exhibited a semi-spherical shape with a high diameter and hydrodynamic sizes as well as negative Zeta potential ($\zeta$) values [47]. Similarly, the sol–gel process and calcination procedure (650 °C) were also employed to obtain magnetic silica-based nanocomposites. This study found that the modification and temperature did not cause significant changes in morphological characteristics, but the calcination decreased the particle size [55].

Negative zeta potential values were observed for mesoporous silica-coated magnetic nanoparticles ($Fe_3O_4$@$mSiO_2$), while the chitosan-wrapping $Fe_3O_4$@$mSiO_2$ nanoparticles exhibited a positive charge (26.94 mV). Furthermore, the hydrodynamic size of the nanoparticles without the chitosan coating was smaller compared to chitosan-wrapping nanoparticles (i.e., the hydrodynamic size increased from 125 to 153 nm with the chitosan wrapping).

In addition, drug loading onto the nanocarrier caused a decrease in the surface area and intensity of XRD peaks [56]. Nanoparticles with a narrow size range were synthesized by co-precipitation and sol–gel methods. Silica-based magnetic nanoparticles exhibited a hydrodynamic size of around 137 nm, a polydispersity index of 0.22, and superparamagnetic behavior. Moreover, silica-coating nanoparticles prevented the oxidation of magnetite to maghemite [57].

Pluronic-F127-coated magnetic silica-based core–shell nanocarriers were synthesized by hydrothermal technique, and the Stöber method was adapted [58]. Nanocarriers presented homogeneous size, spherical, and flower-like, as well as high stability and crystallinity. Furthermore, the polymer percentage affected the particle size (i.e., the average size increased with the Pluronic-F127 content used in the coating process).

Silica-based metal–organic frameworks were synthesized by a sequence of steps, including the sol–gel, hydrothermal, and impregnation techniques [44]. The characterization results show that overall, not only the crystal structure of the MOF was affected by the incorporation of iron oxide but it also caused a decrease in the specific surface area. In addition, IONPs were impregnated into the material structure (pores) in the form of nanoclusters.

Moorthy et al. [27] used solvothermal and sol–gel methods to obtain magnetite nanoparticles and $Fe_3O_4@SiO_2$ nanoparticles, respectively. These nanoparticles were further functionalized with ethylene diamine (EDA) to improve the doxorubicin (Dox) carrying. The textural properties revealed that silica-based nanoparticles exhibit a mesoporous structure and high surface area before the Dox encapsulation. Regarding the average particle size, there was a slight increase after coating with Fuc. Additionally, the SEM images showed a non-regular surface with the presence of aggregates.

The silica-coating Na-doped lanthanum manganites presented a low tendency of agglomeration. The uncoated and coated nanoparticles presented a low coercivity value, which indicates that the materials exhibit ferromagnetic behavior. Nanoparticles also have high colloidal stability in the physiological and tumor environment, allowing their application in biological assays [59].

## 3. Biological and Therapeutic Properties of Silica-Based Nanoparticles

### 3.1. Biomolecule Purification

Microbiological degradation is a common issue that affects the preservation of foods, beverages, and shelf-time. By considering antibiotic resistance, numerous strategies have been developed to prevent the deterioration and damage caused by pathogenic strains [60]. Among these, antimicrobial peptides (AMPs) show excellent activity against different microbial strains. AMPs are the host-defense peptides found in animals, plants, and microorganisms [61]. Although peptides have promising applications, purification and separation processes are difficult and time-consuming as they involve several methods and steps. Additionally, the magnetic separation and immobilization techniques present some advantages, such as simplicity, easy operation, and low cost [50,62]. Recently, Niu and coauthors [63] developed a new strategy (one step) employing silica-decorated $Fe_3O_4$ nanoparticles for AMPs separation. Magnetic nanoparticles were synthesized through the typical co-precipitation method, following silanization and amidation reactions. After the optimization of the purification procedure, it was verified that the adsorption of the AMPs occurred satisfactorily and easily (1 h of contact with the cell-free supernatant).

Furthermore, Benelmekki et al. [64] captured the protein GFP-H6 by adsorption process employing $Ni^{2+}/Co^{2+}$ decorated porous magnetic silica spheres (PMS-Ni and PMS-Co). Subsequentially, from the purification step, it was possible to observe that the capacity of protein capture was enhanced for both doped magnetic nanocomposites. Liu and co-workers [65] synthesized $Cu^{2+}$ modified magnetic mesoporous silica microspheres ($Fe_3O_4@SiO_2Cu^{2+}$) by employing the sol–gel method to capture and enrich the peptides. The results showed that magnetic nanoparticles and $Cu^{2+}$ ions improved the selective peptides and enrichment efficiency due to the affinity between these inorganic compounds and biomolecules.

### 3.2. Food Packaging

The considerable increase in fish consumption and its low shelf-life leads to a substantial need for packaging with properties that make it possible to reduce the degradation caused by deteriorating bacteria, maintaining the flavor and nutritional properties [40]. Essential oils are natural compounds, derivatives of the secondary metabolism of plants with numerous properties and applications [66]. By considering the antimicrobial and antioxidant activities of turmeric essential oil (TEO), Surendhiran et al. [40] synthesized a magnetic silica-based composite with film food packaging for the preservation of Surimi fish. The antimicrobial activity of nanocomposites was evaluated using the *Bacillus cereus* as the deteriorating bacteria model. According to the results, the impregnation of TEO onto magnetic nanocomposite improved bacterial activity due to its slow and controlled release. In addition, the pH of Surimi was maintained, avoiding the oxidation and proliferation of bacteria.

### 3.3. Antimicrobial Activity

Antimicrobial resistance has been considered an emerging health problem due to the ability of microorganisms to resist the pharmacological effect of many drugs [67]. Through gene hypermutation, different pathways may be associated with decreased antimicrobial activity, such as efflux pumps, enzymatic inactivation, decreased drug uptake, and pharmacological target changes (Figure 2) [68]. Thus, the potential advances in nanoscience and nanotechnology have contributed to the development of multifunctional materials that exhibit different properties and applications [69].

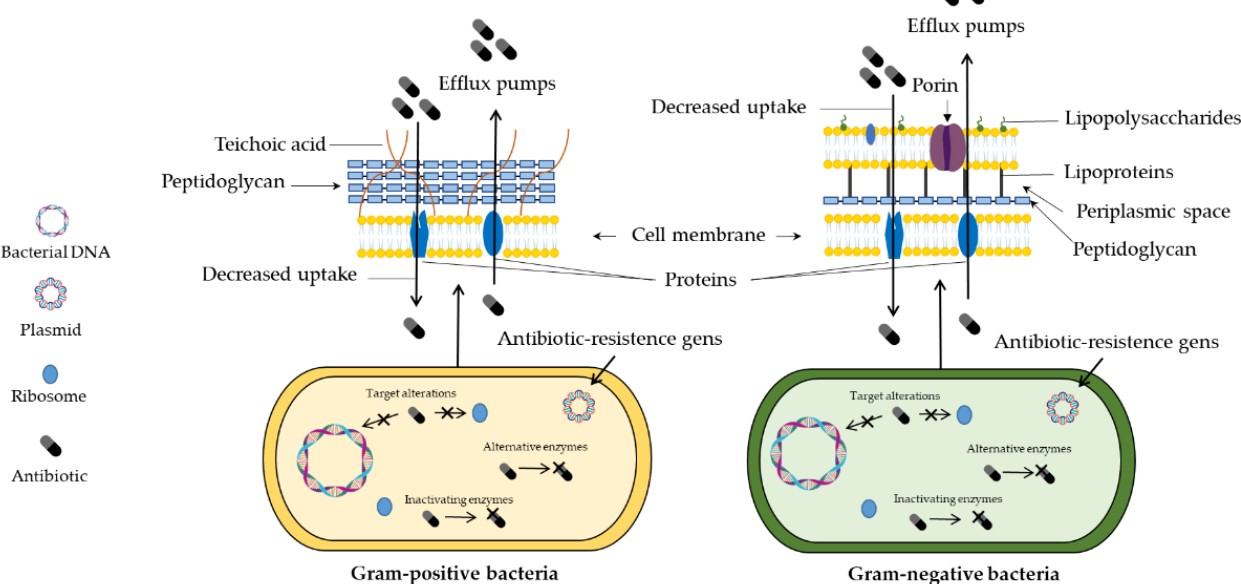

**Figure 2.** Bacterial structure and resistance mechanisms.

The replacement of an anti-inflammatory drug was evaluated by Follmann et al. [41]. Nanohybrids loaded with dexamethasone were used in an antibacterial study to verify their biological activity. The results of the susceptibility assay showed important bactericidal action against *E. coli*, *S. aureus*, *Bacillus subtilis,* and *P. aeruginosa*.

Nayeem et al. [70] developed cationic magnetic nanocomposites containing vancomycin (Van). According to the antimicrobial activity studies, it was verified that the Van impregnated into the nanomaterial showed higher inhibitory capacity compared to the free drug. Similarly, Zhang et al. [71] also immobilized Van onto magnetic hybrid nanoparticles to improve its pharmacological action. Corroborating with Nayeem and coauthors [70], the positive effects observed were the lower dosage to inhibit bacterial growth and the enhanced antibacterial activity.

Chemically modified silica nanoparticles with essential oils also were employed to improve the antibacterial activity. Shahabadi et al. [72] developed magnetic silica nanoparticle-based composites coated with eugenol (EUG). The bacterial activity was evaluated against four food pathogens. The results showed that among the strains tested, *Klebsiella pneumoniae* (Gram-negative) was the most sensitive bacterium. Additionally, the nanoparticles presented higher activity than free essential oil.

Silica nanoparticles present not only high biocompatibility but also exhibit high surface area and reactive sites that enable the conjugation with biocides compounds. Chen and coworkers [53] synthesized core–shell–like ($Fe_3O_4/SiO_2$) magnetic nanoparticles containing a quaternary ammonium salt (QAS) site and three N-halamine sites. The antibacterial assay revealed that the QAS and N-halamine sites act in a synergic manner. Both bacterial strains (*Staphylococcus aureus* and *Escherichia coli*) showed susceptibility to the magnetic compound, with bacterial inactivation of 100% after 30 min of incubations with treatments.

Cationic dendrimers with terminal amine display antimicrobial properties. Nazari et al. [73] also modified the surface of $Fe_3O_4/SiO_2$ nanoparticles to investigate bacterial effects against *S. aureus* and *E. coli*. The inhibitory performance of $Fe_3O_4/SiO_2$ nanoparticles functionalized with dendrimers demonstrated that the minimum inhibitory concentration (MIC) for *E. coli* and *S. aureus* was 4 and 8 $\mu g\ mL^{-1}$, respectively. The higher susceptibility of Gram-negative strain to treatments can be associated with the thickness of the cell wall peptidoglycan.

Transition metals have also been incorporated into the surface of magnetic silica nanoparticles. Shatan and coauthors [51] reported the coating of magnetite with silica nanoparticles and its functionalization with thiol groups to facilitate the impregnation of silver clusters. The antibacterial activity was evaluated using clinical isolates of *S. aureus* and *E. coli* strains. The results clearly express that the decorated $Ag^+$ ions on the surface of the magnetic nanoparticle contribute to the inhibitory effect, considering that no significant difference was observed without the magnetic nanoparticles. In addition, the *E. coli* strain demonstrates higher susceptibility to treatments.

Mesoporous silica-coated magnetic nanocomposites were synthesized for the controlled release of $Ag^+$ ions. Wang et al. [54] reported a novel silica-based core–shell compound for drinking water decontamination. The study to evaluate the antibacterial activity was carried out using *E. coli* as the model. The results showed that the etching time on the nanocomposite structure influenced the disinfectant action.

The copper ions–Schiff base complex was anchored in mesoporous silica-based magnetic nanoparticles. Ahmadi et al. [39] synthesized different magnetic nanoparticles containing streptomycin and investigated the antimicrobial activity against two bacterial strains (*S. aureus* and *E. coli*) in the presence and absence of a magnetic field (H). The results demonstrated that the bacterial strains treated with nanoparticles together with the magnetic field exhibited higher sensitivity to all treatments. Moreover, the nanoparticles containing copper and streptomycin showed the most significant effects. At the same time, the compounds presented antibacterial action more expressively against *E. coli*.

Platinum-doped core–shell magnetic nanoparticles ($Fe_3O_4@SiO_2$-Pt) were developed to evaluate the inhibitory effect against *S. aureus* [42]. The results showed that the antiproliferative effects of nanoparticles occur in a dose-dependent manner. Additionally, the antibacterial activity is improved with the presence of hydrogen peroxide. Similarly, titanium–silica–iron oxide (TSI) nanocomposites were prepared to explore the potential bactericidal activity. The effects of TSI combined with UV light were also studied. The experimental results indicated that the nanocomposites showed bactericidal performance only under the presence of ultraviolet light [43].

The synthesis of a novel magnetic nanocomposite containing different metallic elements and quantities of $Ni^{2+}$ was reported by El Nahrawy et al. [74]. These nanocomposites were tested in diverse bacterial strains. The in vitro data revealed that the compounds with higher proportions of $Ni^{2+}$ ions presented the best inhibitory capacity.

Iron oxide was also impregnated into silica-based microporous materials for antibacterial studies against methicillin-resistant *S. aureus* (MRSA) and *Pseudomonas aeruginosa* [44]. The MIC values and antibiofilm assay showed clearly that the nanocomposite exhibited antibacterial activity and inhibited biofilm formation.

Other magnetic nanoparticles are used to modify the silica surface, such as cobalt ferrites. Babu and Reddy [75] reported the use of $CoFe_2O_4/SiO_2$ core–shell magnetic nanoparticles in an antibacterial study against different bacterial strains. Disk diffusion tests revealed that pure ferrite had no inhibitory effect. In addition, the size and surface area of nanoparticles can influence biological activity due to the higher induction of reactive oxygen species (ROS) release.

Supramolecular assemblies of mesoporous silica were used as a nanoplatform for peptides and antibiotics. Cobalt and manganese ferrites were applied as co-assembly of the system, while melittin and ofloxacin were carried. The in vitro results showed that the AMP and the drug incorporation into the magnetic nanosystem improved the eradication of the antibiofilm. Biofilms are destroyed by the generation of hydroxyl radicals and mechanical forces that guide the active substances into the structure [76].

Nanocomposite containing cobalt and nickel ferrite was recently prepared by El-Sayyad et al. [46]. Antimicrobial and antibiofilm activities were investigated using free compounds and the nanocomposite of interest. Amongst treatments, the nanocomposite exhibited a higher capacity to inhibit microbial growth (fungi and bacteria). Moreover, gamma irradiation improves the antimicrobial activity, but at high doses of $\gamma$-rays, a considerable decrease in the biological activity was observed. Regarding the antibiofilm activity, there was a significant reduction in cell adhesion in the culture of microorganisms submitted to irradiation.

Ehi-Eromosele et al. [59] reported the synthesis of sodium-doped lanthanum manganite coated with silica nanoparticles (LNMO). Different bacterial cultures were used to explore the antibacterial activity of magnetic nanoparticles. Susceptibility assays revealed that pure LNMO nanoparticles exhibited a higher activity than silica-coated LNMO.

Singh et al. [77] investigated a new therapy against *Giardia lamblia* and *Trichomonas vaginalis* using adamantylated organosilatranes. In order to synthesize and design this compound, a simplistic methodology was used based on a covalent anchor of adamantylated silatrane onto the surface of magnetic silica nanoparticles. The results suggested that all the compounds showed an important antiprotozoal action. This product could be considered a non-invasive and controlled drug release system that exhibits remarkable antiprotozoal activity.

*3.4. Drug Delivery and Release Studies*

Although conventional treatments are effective in controlling and curing numerous diseases, pharmacological therapies still present diverse limitations, such as nonspecific action, several side effects, toxicity, and instability to pH and light [78,79]. Thus, nanostructured systems have been widely used to improve the biocompatibility and systemic circulation of drugs (Figure 3) as well as decrease immunogenicity [80].

Some investigations indicated that silica nanoparticles could enhance drug delivery and improve treatment response. In this scenario, Pinna et al. [34] conducted a study using MIL-88A (Fe) MOF crystals nucleated and grown around a polymer core containing superparamagnetic nanoparticles focused on the magnetophoretic drug release of dopamine. The safety profile of MOF was measured using PC12 cells as the neuronal cell model. The findings indicated that dopamine could be delivered into the intracellular compartment using the PMP@MIL-88A carrier as well as avoid side effects.

Moreover, Balasamy et al. [44] reported the action of nanoformulation focusing on biocompatible metal–organic frameworks (MOFs) as well as magnetic nanocarriers for drug delivery and tumor imaging to reveal new cancer therapies. This study developed a nanocomposite comprising Fe/SBA-16 and ZIF-8 (Fe/S-16/ZIF-8) via ultrasonic irradiation.

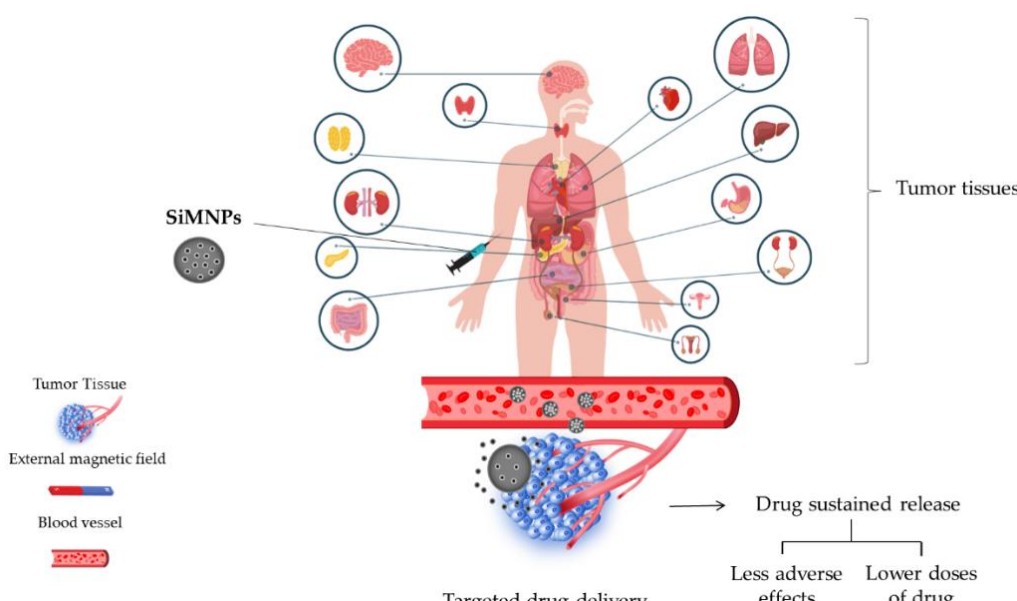

**Figure 3.** Drug delivery systems using silica-based magnetic nanoparticles.

An automated diffusion cell system equipped with a flow-type Franz cell was used analyzed cisplatin (Cis) drug delivery. The results of the drug release profile established the nanocarrier as an excellent platform for cisplatin delivery, enabling a slow and sustained release.

Follmann et al. [41] performed a synthesis of novel hybrid organic–inorganic aerogel materials with one-dimensionally aligned pores to conduct an extended delivery system for a hydrophobic drug. Given the fact that these compounds present hydrophobic pores, the hybrid aerogels exhibited a high drug-loading ability and excellent release pattern for dexamethasone, which was used as a model drug. The results showed a prolonged drug delivery of more than 50 days.

Chitosan-modified magnetic silica nanoparticles and N-isopropylacrylamide (MagSi@Chi-g-NIPAAm) was used as a doxorubicin (Dox) carrier [49]. The drug release study was performed at different pHs (4.0 and 7.4, respectively) to simulate the physiological and tumor environment. In addition, two temperatures were also used to mimic body temperature and hyperthermia conditions (37 and 45 °C), respectively. Regarding the in vitro results, it was verified that the drug release is highly pH/thermos-responsive, i.e., the highest drug release rate was reached at the acidic medium and 45 °C. This behavior is sustained by swelling of the polymers at pH 4.0 (easy protonation of amine groups) and higher rupture of nanoparticles. Similar behavior was observed by Tran et al. [81].

The magnetic nanocarrier containing cisplatin showed higher drug release at pH 5.5 than at pH 7.4. These findings were also attributed to the protonation of $NH_2$ groups in an acidic environment.

Silica-based magnetic nanoparticles were used for targeted delivery and cellular recognition of doxorubicin and miRNA-21 [82]. The effect of the amount of doxorubicin on the loading efficiency (%), as well as the influence of the surface modification with DNA and miRNA-21, was investigated. In vitro results revealed that a gradual increase in the amount of Dox significantly improved drug loading efficiency. Furthermore, the presence of DNA and miRNA-21 in the nanocarrier structure determined the drug release.

Espinoza et al. [58] employed nanocomposite for carrier doxorubicin, MNPs-SiO$_2$-F127-2%, and MNPs-SiO$_2$-F127-4%. The assays were accomplished at two different pHs, 5.4 and 7.4, miming physiological and tumoral conditions, respectively. Drug release data revealed a significant influence on pH. At acidic conditions, the drug release is higher than the pH of 7.4, which may occur by breaking amide bonds. The Pluronic F127 content also affected the drug release rate. Along with this, Wu and co-workers [56] used a nanocom-

posite (Fe$_3$O$_4$@mSiO$_2$ and CS/Fe$_3$O$_4$@mSiO$_2$) to deliver the same drug (Dox) and noted that under acidic conditions (pH 4.0), a higher drug release percentage (%) was reached. Both nanocarriers exhibit a time-dependent behavior; initially, a high concentration was released to the medium, followed by a slower release. The nanocarrier containing chitosan (CS) showed the highest drug release rate due to breaking the "gatekeeper" faster than Fe$_3$O$_4$@mSiO$_2$.

Moorthy et al. [27] studied the fucoidan-coated core–shell magnetic mesoporous silica drug carrier (FeNP@SiOH@Fuc NPs) system action in human breast adenocarcinoma cell line (MDA-MB-231). This system contains a magnetic iron oxide (FeNP) core, a mesoporous silica shell (SiOH), and a marine biopolymer known as fucoidan (Fuc), which focuses on Dox drug delivery involving pH and hyperthermia.

In a similar manner, a novel Dox nanocarrier based on nanoparticles loaded into multi-walled carbon nanotubes with mesoporous silica (mSiO$_2$) MWCNTs@CoFe$_2$O$_4$@mSiO$_2$ was developed. The results showed a remarkable pH-responsive drug delivery character within 48 h, targeting cancer therapy improvement [83]. Wang et al. [84] produced Fe$_2$B@SiO$_2$ nanoparticles covalently grafted from graphene oxide containing 90% entrapped epirubicin (EPI). The drug delivery was pH-dependent, with higher EPI release achieved at pH 5.7.

Wang et al. [45] investigated the synergistic antitumor effect of the hydrogel-based nanocomposite (MMSN-RBITCs) for Dox delivery by MTT assay. The free Dox showed a low cytotoxicity capacity for the 4T1 cell line over 24 h of contact. In contrast, the Dox-loaded hydrogel was able to decrease cell viability. Moreover, the cytotoxicity increased dramatically when the hydrogel was exposed to magnetic hyperthermia therapy (MHT), indicating that heat potentiates the cytotoxic effects of nanoparticles.

In a study reported by Solak et al. [85], disulfiram loading on mesoporous silica magnetic nanoparticles modified with acid-linked polyethylenimine (mMDPF) and in vitro drug release was shown. At slightly acidic conditions (pH 6.0), the drug was released in a faster and more controlled manner than under physiological conditions (pH 7.4). These results demonstrate a promising use of the nanoparticles as a nanoplatform for the release of disulfiram, considering the faster release in an acidic environment, avoiding the drug inactivation by the P-glycoprotein.

Laranjeira et al. [57] synthesized the slow release of exemestane through a nanodevice for intravenous administration. The encapsulation efficiency and drug loading capacity were 90% and 37.7%, respectively. The drug release time was over 72 h, which indicates the exemestane was adsorbed only in the inner pore walls of the nanoparticles. These results show the promisor application of silica-coated mesoporous magnetic nanoparticles to increase drug release.

Recently, a novel dehydropeptide hydrogelator, Npx-L-Met-Z-ΔPhe-OH, was produced and associated with two different plasmonic/magnetic nanoparticle architectures, such as core/shell manganese ferrite/gold nanoparticles and gold-decorated manganese ferrite nanoparticles, targeting cancer therapy. This complex was tested using a model antitumor drug, curcumin, a natural bioactive molecule revealing promising results for photothermal and photo-triggered drug delivery [86].

### 3.5. Antioxidant Activity

Oxidative stress is known as an organism imbalance between the production of oxidative substances and antioxidant compounds [87]. Free radicals are atoms or molecules that have unpaired electrons in the valence shell. This characteristic enables those free radicals the ability to react with macromolecules (Figure 4) and trigger several diseases [88].

Mesoporous silica nanoparticles are widely used in biological studies due to their distinct properties. A recent report showed that quercetin conjugated with silica-based magnetic nanoparticles was able to significantly reduce Aβ40-induced ROS generation in primary hippocampal neurons [89].

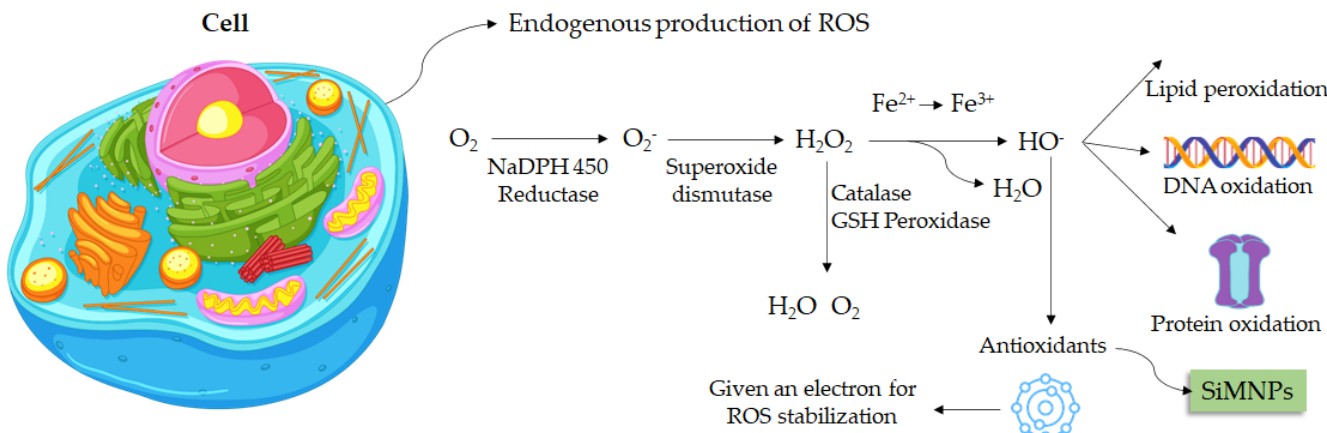

**Figure 4.** Silica-based magnetic nanoparticles focusing on antioxidant activity.

The antioxidant activity of silica-based magnetic nanoparticles functionalized with chitosan and tannic acid ($\gamma$-Fe$_2$O$_3$@SiO$_2$-CS-TA) and also amine groups ($\gamma$-Fe$_2$O$_3$@SiO$_2$-NH$_2$-TA) was reported by Świętek et al. [90]. DPPH assay results showed that nanoparticles modified with chitosan exhibit a slightly higher antioxidant capacity. At the same time, the amount of TA incorporated into the magnetic nanoparticles and the concentration of treatments also influenced the ability to scavenge free radicals.

Zare and Sarkati [48] synthesized iron oxide nanoparticles coated with silica and chitosan immobilized into the surface (Cs-f-SiO$_2$@Fe$_3$O$_4$ MNPs). Magnetic nanoparticles were used as a carrier for silymarin. The antioxidant activity assay showed that silymarin incorporated into magnetic nanoparticles presented higher DPPH radical scavenging ability compared to pure nanoparticles. Corroborating with results reported by Świętek et al. [89], the inhibitory capacity increases with the concentration of compounds.

Patsula et al. [91] investigated the antioxidant activity of $\gamma$-Fe$_2$O$_3$@SiO$_2$-PEI-PEG nanoparticles coated with poly(l-lysine) (PLL) and polyethyleneimine (PEI) + PLL (PLLL). The antioxidant power of magnetic nanoparticles at the cell level was investigated by flow cytometry. According to the results, the treatments showed the capacity to reduce the endogenous production of ROS in human glioblastoma cells (U87MG cell line).

*3.6. Silica-Based Nanoparticles Biocompatibility and Safe Profile Assessment*

Investigations that focus on discovering new therapeutic agents that exhibit low side effects are significant to the nanomedicine field. In this scenario, studies that analyze the biocompatibility of these compounds are essential. Keshavarz et al. [35] reported that silica nanoparticles improved drug release using Tamoxifen (anticancer treatment) by hyperthermia. This investigation used poly (carboxybetaine methacrylate) (pCBMA) coating for magnetite mesoporous silica nanoparticles (MMSNs) to inhibit protein uptake and avoid the protein corona effect. The findings of this study indicate that nanogels are biocompatible and present no cytotoxic effects since it was tested on L929 cells at different concentrations and distinct treatment times of 24, 48, and 72 h.

Ledda et al. [36] assessed the biocompatibility of sub-5 nm silica-coated superparamagnetic iron oxide nanoparticles in human stem cells and mice for possible use in nanomedicine therapies. The results showed that this system did not alter any of the stem cells' tested parameters, such as growth, viability, morphology, cytoskeletal organization, cell cycle progression, immunophenotype, and the expression of pro-angiogenic and immunoregulatory paracrine factors. Neither the osteogenic nor myogenic differentiation markers suffered any modification. Moreover, in vivo analyses using a mice model suggested no acute or chronic cytotoxicity, exhibiting short- and long-term biocompatibility and safe profile via analyses of histopathology, hematology, serum pro-inflammatory response, body weight, and mortality.

The biocompatibility and bioaccumulation of nanoparticles were investigated by employing mice as an animal model. The use of magnetic nanoporous silica nanoparticles (MNPSNPs) in an in vivo study was reported by Janßen et al. [92]. By considering the difficulty for magnetic nanoparticles to accumulate in deeper regions, a magnetic implant was inserted subcutaneously to improve circulation. The analysis of biological tissues showed that the nanoparticles accumulated mainly in the lung, liver, and spleen. However, depletion occurred quickly, and no significant tissue changes were observed. In a similar manner, Nasiri et al. [93] also reported that silica-coated iron oxide nanoparticles caused insignificant changes in histological tissues.

Mesoporous silica nanoparticles with magnetic core ($Fe_3O_4$) coated with polyethylene glycol-phospholipids were used in in vitro and in vivo studies to evaluate biocompatibility. The cellular uptake was explored using hepatic cells as the model (HepG2 and HepaRG cells). There was aggregation due to its high ionic strength, protein adsorption, and accumulation in the liver and spleen. Cell internalization was slower with coating nanoparticles than with native ones. Despite the accumulation, the nanoparticles did not cause toxicity in these tissues [94].

Navarro-Palomares et al. [37] reported that dye-doped biodegradable nanoparticle $SiO_2$ coating on zinc- and iron-oxide nanoparticles enhanced biocompatibility and can be used for in vivo imaging investigations via supplying luminescent functionality to zinc and iron oxide multifunctional nanoparticles.

With the aim of increasing biocompatibility and avoiding protein adsorption and ion aggregation in the biological environment, SiMNPs were conjugated with PEG (MFO/$SiO_2$-PEG). The cytotoxicity of MFO/$SiO_2$-PEG was investigated in NIH/3T3 cell line (MFO/$SiO_2$-PEG concentration from 2 $\mu g \cdot mL^{-1}$ to 200 $\mu g \cdot mL^{-1}$) by CCK-8 assay. In this range, the complex showed high biocompatibility (cell viability remained above 80% for all treatments) [95].

Similarly, Laranjeira et al. [57] investigated the cytotoxicity of SiMNPs and MNPs in Human Gingival Fibroblasts (HGnF) cells. The results showed changes in metabolic activity and a decrease in cell viability by around 30%, which indicates that nanoparticles are biocompatible. In addition, no changes in cell morphology were observed after 24 h of incubation. Follmann et al. [41] also reported a high biocompatible of multifunctional hybrid aerogels containing silica nanoparticles, assessed on Vero cells and L929 fibroblasts.

The microwave-assisted synthesis method is a promising alternative to produce magnetic nanocomposites under mild conditions and eco-friendly approaches. Gharibshahian and co-authors [52] developed superparamagnetic mesoporous silica-coated cobalt ferrite nanoparticles ($Co_2Fe_2O_4/SiO_2$) using microwave-modified Pechini technique, while the coating with silica was carried out by the hydrothermal method. The cytotoxic effects evaluated in 3T3 fibroblast cells showed that nanoparticles caused a significant decrease in cell viability only at the highest concentration (500 $\mu g$ $mL^{-1}$). Previous reports correlate the toxicity of nanoparticles with the release of metal ions at the high concentrations tested [96]. The higher toxicity of $Co_2Fe_2O_4$ synthesized by microwave radiation can be attributed to energy absorption, which contributes to cell damage. In addition, the coating of magnetic nanoparticles improves biocompatibility.

Calcium-doped silica-coated magnetic nanoparticles were used to verify the in vitro cytotoxicity of viral complexes [16]. The cytocompatibility of the nanocarriers was determined by the MTT assay using the MS1 cell line (pancreatic islet endothelial cells). The in vitro results revealed that pristine IONPs and silica-coated IONPs have good biocompatibility only at low concentrations (20 $\mu g$ $mL^{-1}$). Moreover, the hydroxide calcium-doped nanoparticles ($Ca(OH)_2$) have more influence on necrotic effects, while calcium citrate-doped NPs are involved with apoptosis. Regarding the MNPs functionalized with lentiviral vectors, the experimental data indicate that the type of nanoparticle (IONPs, silica-coated IONPs, calcium source) determines the time of internalization. Among magnetic nanoparticles, magnetite-based carriers exhibited the highest cell transduction capacity.

Meantime, the in vivo results showed that organs submitted to the magnetic field exhibited a high expression of GFP compared to the group non-exposure to the external magnet. The spleen was the most source of iron bioaccumulation. The intratumoral retention images indicate that the type of application (intratumoral or intravenous) of the magnetic field presence exerts a significant effect on the biodistribution/accumulation of MNPs.

On the other hand, Chakkarapani et al. [97] investigated the internalization of silica-coated magnetic nanoparticles as well as the intracellular trafficking probed with rhodamine B in HEK293, NIH3T3, and RAW 267.4 cell lines. Macrophages were more susceptible to these nanoparticles than kidney cells or fibroblasts. The cellular uptake was high for the RAW 267.4 cells compared to other cell lines. Additionally, it was possible to observe the presence of micronuclei, which is related to genotoxic effects.

The cytotoxic effect of silica-coated magnetic nanoparticles containing rhodamine B was evaluated in BV2 microglial cells [98]. Nanoparticles induced the production of ROS and accumulated in the cells, activating the microglial cells, which are involved in immune homeostasis at a concentration of 0.1 µg µL$^{-1}$. At the same time, the MNPs, decreased the glucose uptake by BV2 cells in a dose-dependent manner. Once combined with glutathione and citrate, a significant reduction in nanotoxicity and penetration to the brain was observed, protecting the hippocampus and thalamus.

Shin et al. [99] reported the lowering of the toxic effects induced by rhodamine B functionalized silica-coated magnetic nanoparticles mediated by glutathione and citrate. The results confirmed the BV2 activation, ROS, and reactive nitrogen species (RNS) secretion, nanoparticle lysosomal accumulation, and vesicle formation. At 100 µg mL$^{-1}$, the gene expression related to metabolism and inflammation was altered. Additionally, microglial cells enhanced serine secretion in the culture medium and reduced the ATP concentration as well as the neuron's proteasome activities. Therefore, microglial activation was associated with D-serine secretion.

### 3.7. Silica Nanoparticles Carriers Focusing on Cardiovascular Treatment

Huang et al. [100] used silica nanoparticles to improve stroke treatment. This study was conducted using a novel surface imprinted magnetic mesoporous silica nNOS–PSD-95 (nitric oxide synthase–postsynaptic 12 density protein-95) as artificial antibodies. This complex showed a significant adsorbing performance and high recyclability and can be used as a sorbent to catch uncouplers from natural compounds. Biological action and interaction were measured in vitro and in vivo. This compound showed neuroprotective action on glutamate-injured PC12 cells (rat adrenal medulla cell line) and uncoupling activity targeting nNOS–PSD-95. Moreover, nNOS–PSD-95 uncouplers improving neurological deficit and decreased infarct volume of MCAO/R (middle cerebral artery occlusion and reperfusion) in rats, indicating significant potential for ischemic stroke treatment.

### 3.8. Silica Nanoparticles against Cancer Cells

The development of nanoparticles with potential anticancer activity is a very important aspect to be investigated, considering the chemoresistance, recurrence, and several side effects [101]. Previous investigations reported that silica nanoparticles present action against different cancer cell lines [27,28,30,31]. In addition, silica nanoparticles are widely explored due to their inert characteristic and high biocompatibility (Figure 5). Moreover, they can be easily obtained from sustainable sources such as rice husks by a top-down approach [24]. Chemically modified silica nanoparticles are synthesized for carrier drugs and other organic compounds aiming to explore their antitumor activity. Due to responsiveness to an external magnetic field and biocompatibility, magnetic nanoparticles are commonly used in modification reactions [71].

In this scenario, Abbasi Kajani et al. [47] synthesized gold/silver decorated silica-coated iron oxide nanoparticles and evaluated the anticancer activity using human breast (MCF-7) and cervical cancer (HeLa) cell lines. According to the results, it was possible to

verify a dose-dependent anticancer activity. In addition, higher effects were exhibited on the MCF-7 cell line than on HeLa. The nanostructures containing gold/silver showed high results when compared to silica–iron oxide nanoparticles (FeSi-NPs), indicating that metal ions improve the cytotoxicity against tumor cells. Moreover, nanoparticles showed good biocompatibility with healthy cells, significantly affecting cell viability only at the highest concentrations.

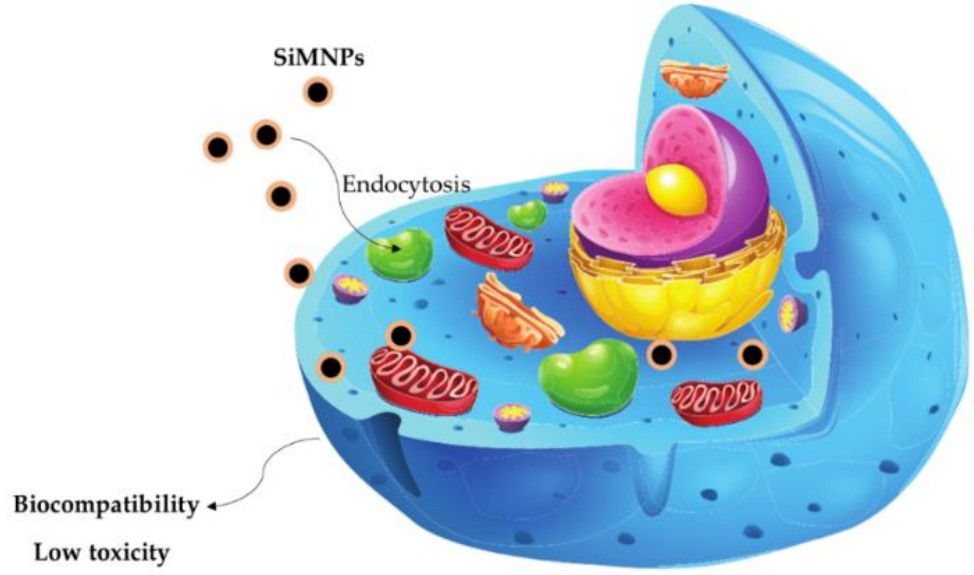

**Figure 5.** Interaction of silica-based nanoparticles with healthy cells.

Taher et al. [102] synthesized magnetosome mimics nanoparticles with a lipid-covered magnetite core synthesized by the bacterium *Magnetospirillum magneticum* (called magnetosome). Cytotoxic effects of mimetic magnetosomes were evaluated using MDA-MB-231 (human breast adenocarcinoma) cells by flow cytometry. In vitro results showed that magnetosomes exhibited higher toxicity than nanoparticles. Among them, the silica-containing mimetic magnetosomes (Si@rMNP) showed a higher capacity for cellular internalization.

Iron oxide-loaded hollow silica spheres were synthesized and evaluated by MTT on HCT-166 cells (colon-rectal carcinoma). The cellular viability decreased in a dose and iron-dependent manner. These data suggest that the coupling with magnetic nanoparticles could increase the anticancer capacity of the iron oxide-loaded hollow silica spheres [55].

Foglia et al. [14] developed silica-coated magnetic iron oxide fluorescent nanoparticles. Confocal microscopy confirmed internalization in human colon cancer cells (Caco-2). However, no interference with cytoskeleton organization or cell cycle progression was observed.

Solak and coauthors [85] employed magnetic mesoporous silica nanoparticles modified with acid-linked polyethyleneimine (mMDPF) to loaded Disulfiram (DSF) against MCF-7 and MCF-10A cell lines. The results of cytoviability showed that MCF-7 cells were the most sensitive line to treatments. The higher selectivity of nanoparticles modified with folic acid is attributed to the expression of fatty acid (FA) receptors by MCF-7 cells [103].

The combined therapy of $Cu^{2+}$ ions and/or sodium nitroprusside (SNP) improved the cytotoxicity of the nanoparticle-loaded DSF, although it did not enhance the effects of free disulfiram. Further, the magnetic field also contributed to higher cellular uptake and drug release.

Itatahine et al. [30] reported that multifunctional carbon nanomaterials focus on anticancer drug delivery to improve cancer therapies and decrease side effects. This study analyzed a multifunctional hybrid nanomaterial composed of magnetic graphene oxide (GO) and magnetic carbon nanotubes (CNTs) coated with mesoporous silica. The system was used to load and deliver an anticancer drug, camptothecin (CPT). The findings showed

that this complex is more effective against the HeLa cell line when compared to CPT isolated. Graphene oxide was also used to synthesize silica-based nanoparticles ($Fe_2B@SiO_2@GO$) as an epirubicin carrier. The toxicity against HeLa cells revealed dose-dependent effects [84].

Silica-coated iron oxide nanoparticles were conjugated with papain as antitumoral to HeLa cells. This system could generate apoptosis in HeLa and HSF 1184 cells by condensation and fragmentation of chromatin. The hemocompatibility results showed that silica-coating protected against the toxic effects of pristine magnetic nanoparticles [93].

The safety profile and antitumor activity of free silica-based magnetic nanoparticles containing cisplatin were tested [44]. The cytotoxicity activity of Fe/S-16/ZIF-8 was analyzed in MCF-7, HeLa cells, and Human Foreskin Fibroblast (HFF-1) cells. The results showed an important synergistic effect of cisplatin drug delivery found in Fe/S-16/ZIF-8. The lethal concentration ($LC_{50}$) of Fe/SBA-16/ZIF-8 for MCF-7 and HeLa cells was estimated at $0.119$ mg mL$^{-1}$ and $0.028$ mg mL$^{-1}$ at 24 h, respectively. While for HFF-1 cells, the $LC_{50}$ was $0.016$ mg mL$^{-1}$.

Ferreira et al. [104] developed a nanocarrier (core–shell silica-coated iron oxide NPs functionalized with dicarboxylic acid groups, called $Fe_3O_4@SiO_2$-TESPSA (MSF)) conjugated with cisplatin. The cytotoxicity assays indicate that the free drug was more toxic against healthy cells (human pancreatic duct epithelial-H6C7) than malignant cells. However, the cisplatin conjugation improved the biocompatibility with normal cells, demonstrating that the nanoparticles increase the drug specificity. Corroborating with MTT results, antiproliferative assay revealed more significant effects against tumor cells with a high metastatic rate (AsPC-1) in comparison with the tumor line with a low metastatic rate and healthy cells. These results can be attributed to the presence of TESPA (dicarboxylic acid groups), which improves the antitumor activity.

Tran et al. [81] developed magnetic mesoporous silica nanoparticles (core–shell) conjugated with a fluorescent probe for cisplatin carriers. The nanoparticles were internalized by HeLa and SHSY5Y, showing low cytotoxicity in these cell lines. The transporter system was not toxic to HEK293 cells (human embryonic kidney epithelial cells). Furthermore, nanoparticles destroyed HeLa cells under near-infrared (NIR) at 808 nm by chemical-photothermal effect.

A study analyzed a novel anticancer therapy based on tumor starvation focused on synergism to improve chemodynamic therapy and stimulated chemotherapy [105]. In this scenario, mesoporous silica nanoparticles (MSN) and glucose oxidase (GOX) were grafted onto its surface, and $Fe_3O_4$ nanoparticles (Fenton reaction catalyst) and a hypoxic prodrug of tirapazamine (TPZ) were loaded in this system. The results indicated that this complex could be an excellent anticancer therapy because GOX can decrease cancer cells' nutrients and increase ROS production, especially by the Fenton reaction (Figure 6). Wang et al. [45] reported that the oxidative damage caused by Rhodamine B isothiocyanate (RBITC)-labeled mesoporous silica nanospheres with Mn-Zn Ferrite ($Mn_{0.6}Zn_{0.4} Fe_2O_4$) nanoparticle core (MMSN-RBITC) in 4T1 cells occurs through the conversion of endogenous $H_2O_2$ in ·OH radicals (Figure 6). Cell morphological changes and a decrease in tumor growth were observed in an in vivo study. However, nanoparticles showed excellent biocompatibility. The presence of an alternate magnetic field improved the toxic effects on the tumor line. The heat generation by MHT could increase the amount of ·OH in the intracellular environment accelerating cellular apoptosis.

Duan et al. [106] produced self-assembled multifunctional nanohybrids with a controllable disassembly capacity to improve anticancer therapy. These nanohybrids presented high gene transfection capacity and minimal cellular toxicity (HepG2 and C6 cell lines). In addition, in vivo results demonstrated that the nanoparticles were safe and biocompatible with the animal model. Moreover, this complex exhibits photothermal activity that could improve photothermal-based treatments.

Cytotoxic effects induced by the Fenton reaction and ROS were also reported by Qin et al. [107]. This work highlighted the importance of $Fe_3O_4@SiO_2$ mesoporous spheres as Fe(ii) donors loaded with artemisinin (ART) and a photosensitizer to improve tu-

mor hypoxia in photodynamic therapy (PDT) for targeting antitumor treatment. ART was able to increase ROS production by a pH-mediated Fenton-like process. Moreover, this investigation reported that NIR light irradiation could raise ROS generation and improve antitumor treatment. The increase in reactive oxygen species allows the accumulation of lipid peroxides, resulting in a type of programmed cell death called ferroptosis (Figure 6) [108].

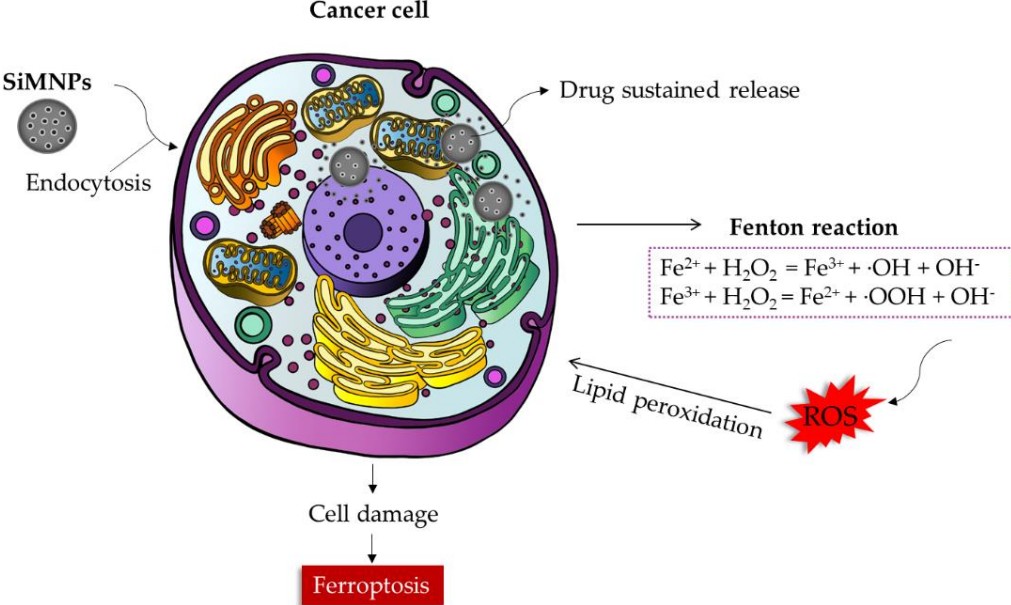

**Figure 6.** Silica-based magnetic nanoparticles enhance antitumor activity.

Asghar et al. [109] conducted a study based on a smart nanocarrier with a mesoporous magnetic core and thermo-responsive shell for the co-delivery of doxorubicin and curcumin to improve anticancer therapy. The findings showed that in vitro studies using normal and HeLa cells showed the non-toxic nature of the MIOP(NIPAM-Mam) nanocomposite. On the other hand, the co-loaded MIOP(NIPAM-Mam)-Cur-Dox nanocomposite presented a significant anticancer activity when compared to free Dox, free Cur, and a free Dox/free Cur mixture. Thus, this system presents a remarkable potential for directed and targeted drug release for cancer therapy.

Moreover, Yang et al. [110] developed a novel biodegradable magnetite/doxorubicin ($Fe_3O_4$/Dox) co-loaded PEGylated organosilica micelles (designated as FDPOMs). Nanoparticles present high circulating stability and excellent GSH-triggered biodegradability to emphasize magnetically targeted magnetic resonance imaging (MRI) and cancer treatment. The results indicate that this system showed a distinctive GSH-responsive release and selective cytotoxicity effect against cancer cells. Liu et al. [65] also developed a biocompatible nanocarrier to decrease the non-selective action of doxorubicin. The results showed that Dox loaded into a magnetic nanosystem exhibited higher toxicity against tumor cells (HepG2) than healthy cells (HL-7702).

Gao et al. [31] developed a new theranostic drug release complex focused on radial mesoporous silica, that is, hybridized with multiscale magnetic nanoparticles for MRI-guided and different magnetic fields (AMF) to improve breast cancer treatment. Doxorubicin, a drug frequently used to treat breast cancer, was loaded into the mesochannels.

The results found in this study indicate excellent drug loading and AMF stimuli-responsive release capacity. Analyses in vitro and in vivo s were performed to investigate the safety profile of this system and its anticancer action. The findings indicate that these complexes may be applied as a drug delivery system since they present a selective effect, acting against the breast cancer cell line (MCF-7) and exhibiting low effects in the mouse fibroblast cell line (L929). Moreover, in vivo anticancer analyzes indicate that this complex

presents an excellent capacity to be used as an MRI-guided stimuli-responsive drug delivery system to improve breast cancer treatments.

Magnetic silica-based nanoparticles also were synthesized by Espinoza and coauthors [58]. Nanocomposites containing iron oxide nanoparticles, $SiO_2$, and Pluronic F127 (MNPs-$SiO_2$-F127-2% and MNPs-$SiO_2$-F127-4%) were used to carrier doxorubicin. The safety profile and cytotoxicity of nanocarriers were performed using the healthy line (human embryonic kidney cells-HEK293T) and tumor line (liver cancer-HepG2). The biocompatibility of magnetic nanoparticles was verified through the (3-(4,5-dimethylthiazol-2-yl)-2,5-diphenyltetrazolium bromide) assay (MTT). In vitro assays revealed that both pure MNPs and synthesized nanocarriers were cytocompatible with cell lines. The antitumor activity data of Dox loaded on nanoparticles indicate that effects were potentiated with incubation time and concentration. Wu et al. [56] synthesized a mesoporous magnetic nanocomposite wrapped with chitosan (CS/$Fe_3O_4$@$mSiO_2$) to control Dox release. Similarly, pure nanocarrier ($Fe_3O_4$@$mSiO_2$) presents an insignificant decrease in cell viability against HepG2, while CS/$Fe_3O_4$@$mSiO_2$-Dox shows an effect in a dose-dependent manner. Additionally, the higher cytotoxicity of Dox loaded into CS/$Fe_3O_4$@$mSiO_2$, in comparison to the free drug, can be attributed to the oxygen capture by $Fe^{3+}$ and $Fe^{2+}$ from the cells [111].

Mesoporous silica-based nanoparticles functionalized with aptamers were used by Sakhtianchi et al. [112] for doxorubicin loading. The encapsulation results showed an efficiency of 67%. The MTT assay demonstrated that decorated particles showed higher cytotoxicity in MCF-7 cells than without decoration. The cellular uptake was observed by confocal microscopy and was also higher for decorated ones.

Oltolina et al. [113] developed a monoclonal antibody-functionalized iron oxide as a carrier for doxorubicin. The effect of silica or calcium coating on cytocompatibility was evaluated. The antibody functionalization allowed the nanoparticle to interact with the cell surface without internalization. However, nanoparticles were able to deliver drugs, reaching cellular nuclei (GTL-16 and Huh7 cells) as well as causing cytotoxicity, by MTT assay, after 72 h of incubation with concentrations varying from 6.25 to 50 µg mL$^{-1}$. The calcium coating showed a significant level of apoptosis and necrosis (around 20%), while the silica-coated nanoparticles promoted a slight reduction in cell viability (around 80%).

Pilapong et al. [114] evaluated the cell viability and cell cycle disruption in hepatocellular carcinoma (HepG2) cells and peripheral blood mononuclear cells (PBMC) after contact with SiMNPs and SiMNPs conjugated with EpCAM aptamer-conjugated $SiO_2$-MNP (EpCAM-mMNPs). The results showed that the nanoparticles did not affect cellular proliferation, which indicates their biocompatibility with these cell lines. The EpCAM-mMNPs cellular uptake was investigated in cell EpCAM+ (HepG2 and K562/ADR), cells with moderate expression (K562), and cells EpCAM-(PBMC), confirming the high affinity of EpCAM-mMNPs with EpCAM+ cells. In this study, the cellular uptake pathway occurred by endocytosis. Moreover, images obtained by electron microscopy analysis confirmed the presence of EpCAM-mMNPs within intracellular compartments, including lysosome/endosome.

Maboudi et al. [115] developed a nanocomposite (BSA shell-BSA-MS) using bovine serum albumin (BSA) and magnetite nanoparticles cluster core (MS) to evaluate the cytotoxicity against the HepG2 cells. The hybrid nanoparticles present a slight effect on cell viability only at the highest concentration (100 µg mL$^{-1}$), while MS particles showed cytotoxicity activity up to 50 µg mL$^{-1}$. From the results expressed by the lactate dehydrogenase (LDH) assay, there was no statistically significant increase in the release of the LDH enzyme, suggesting that the materials did not cause the rupture of the plasma membrane. The functionalization of nanoparticles with albumin improved silica cytocompatibility [116].

A previous study reported a remarkable action of multicomponent nanoparticles, consisting of an iron oxide core and a mesoporous silica shell, against glioblastoma cells. This investigation showed the anticancer effect in glioma cells by delivering drugs across

the blood–brain barrier as well as combining with the exposition of alternating low-power radiofrequency (RF) fields [117].

Surface-functionalized silica-based nanoparticles have been used to improve biological activity. The cellular uptake of $\gamma$-Fe$_2$O$_3$@SiO$_2$-CS-TA and $\gamma$-Fe$_2$O$_3$@SiO$_2$-NH$_2$-TA was determined using the LN-229 cell line (parieto-occipital glioblastoma) in the presence and absence of magnetic field. From the experimental data, it was found that variables such as the amount of tannic acid, magnetic field, the concentration of treatments, and type of chemical modification caused significant changes in cellular uptake [90].

Li et al. [118] analyzed a sequential treatment for pancreatic cancer via losartan and gemcitabine-loaded magnetic mesoporous spheres. This study was conducted using amine-functionalized Fe$_3$O$_4$ embedded periodic mesoporous organosilica spheres (Fe$_3$O$_4$@PMO-NH$_2$) and Fe$_3$O$_4$@PMO as drug delivery. Type I collagen and hyaluronic acid present in the pancreatic cancer microenvironment can be inhibited by losartan. The tumor volume showed a significant reduction when treated with this complex, suggesting that this system was able to improve cancer treatment.

Another field very important to reveal novel anticancer therapies is based on natural products that present several bioactive molecules, presenting different biological proprieties, such as anticancer, that frequently exhibit excellent safety profiles. The antiproliferative effects of Cs-f-SiO$_2$@Fe$_3$O$_4$ MNPs and silymarin carried (SIL-Cs-f-SiO$_2$@Fe$_3$O$_4$ MNPs were investigated against the breast cells (MCF-7). As reported by Zare and Sarkati [48], magnetic nanoparticles containing silymarin showed higher cytotoxicity than the pristine compound. Moreover, the toxic effects occurred in a dose-dependent manner.

Taking this into account, Xing et al. [119] investigated Janus nanocarriers for magnetically targeted and hyperthermia-enhanced curcumin liver cancer treatment. The results of this study indicated that Janus M-MSNs-Cur presented an important anticancer action and a remarkable safe and selective profile when analyzed in vitro and in vivo models.

Jin and coauthors [120] synthesized mesoporous silica doped with Gadolinium (Gd) (GdMSNs) coated with liposome to load KLA peptide, called KLA@GdMSNs-LP. The cell internalization results showed that the KLA peptide carried into the liposome has higher uptake due to the double-layer lipidic allowing the interaction with the cell membrane. Whereas GdMSNs displayed high hemotoxicity and cytotoxicity, the liposomes containing only GdMSNs did not exhibit significant effects. At the same time, the MTT assay indicated that the KLA cytotoxicity and apoptotic effects were improved with the loading onto liposomes.

Iliasov et al. [121] evaluated the influence of MNPs coating with silica in an in vitro cytotoxicity study. The cell viability results showed that nanoparticles presented toxic effects on the cell line (PC3-human prostate cancer) only at higher concentrations. At the same time, an increase in cellular uptake of nanoparticles was observed with preincubation time. However, the cytotoxicity cannot be attributed to coating, considering that pure MNPs exhibited a higher cellular uptake than core–shell nanoparticles. The damage to the cell membrane measured by lactate dehydrogenase assay demonstrated that silica-coated MNPs caused a significant release of cell components. Moreover, cell death is associated with necrosis due to an intracellular increase in Ca$^{2+}$.

Nanoferrites are used in biological studies due to their high chemical stability, anisotropy, and saturation magnetization. Sichamnan et al. [122] produced magneto-fluorescence nanoparticles (with carbon dots and silica-coated cobalt-manganese nanoferrites). The system was developed to be applied as a potential fluorescent sensor and magnetic hyperthermia. In vitro and in vivo results showed low cytotoxicity in HeLa cells and zebrafish.

Similarly, Kavkhani et al. [25] developed a hybrid nanocarrier platform for cell image, diagnosis, and therapy using mesoporous silica-coated MNPs (MnFe$_2$O$_4$@mSiO$_2$). The in vitro cytotoxicity of MnFe$_2$O$_4$@mSiO$_2$ and MnFe$_2$O$_4$ was investigated using the A549 cell line (lung cancer). The effects of nanoparticles on cell viability and antiproliferative activity were measured by the MTT assay. The results showed that MnFe$_2$O$_4$@mSiO$_2$ is more biocompatible than MnFe$_2$O$_4$ due to the protection against acid leaching. The

cell viability was also maintained above 80% at all concentrations tested for 24 and 48 h. However, at 72 h, a significant decrease in cell viability was observed in a dose-dependent manner. Corroborating the MTT results, the 4′,6-diamidino-2-phenylindole assay (DAPI) revealed that the nanoparticles did not cause cell changes within 24–48 h but induced cell death over long periods (72 h).

Khanna and co-workers [10] evaluated the toxicity of copper ferrites ($CuFe_2O_4$) synthesized at different temperatures (300, 500, and 700 °C) designated as Cu3, Cu5, and Cu7, respectively. The cytocompatibility assay revealed that magnetic nanoparticles were safe, maintaining cell viability of around 80%. By considering their best magnetic properties, Cu3 nanoparticles were chosen for the silica coating. The in vitro cytotoxicity assays were performed using the J77 cell line. As reported, silica-coated copper ferrite nanoparticles (SiCu3) presented higher compatibility than pure copper nanoparticles, even at high concentrations. Cell flow cytometry was employed to analyze the effect of nanoparticles on cell cycle progression. Both nanoparticles (Cu3 and SiCu3) caused an increase in cell viability at all concentrations (25–250 $\mu g\ mL^{-1}$). The protective behavior of silica avoids acid erosions, decreases accumulation, and improves intracellular uptake [123].

### 3.9. Point-of-Care Detection of Silica-Based Nanoparticles

Silica nanoparticles are highlighted in point-of-care detection to enhance imaging assessment and improve the diagnosis in nanomedicine [37,38,124]. In this scenario, a recent review summarized the importance of silica nanoparticles for improving the sensitivity of lateral flow assays (LFAs), which is a rapid and affordable point-of-care device for health diagnostics. Silica nanomaterials have been used to enhance the signal of label materials or offer stability and subsequent detection performance improvement [125]. Moreover, Monaco et al. [38] developed a smart assembly of Mn-ferrites/silica core–shell with fluorescein and gold nanorods that exhibit robust and stable nanomicelles for in vivo triple modality imaging. The results of this study indicated a contrast-to-noise ratio (CNR) as high as 60 in a mouse leg injected subcutaneously with the nanosystem. Moreover, biocompatibility analysis did not show hemolytic effects, highlighting this investigation's medical diagnosis. Additionally, Navarro-Palomares et al. [37] developed a dye-doped biodegradable nanoparticle $SiO_2$ coating on zinc- and iron-oxide nanoparticles to enhance biocompatibility and in vivo imaging investigations. In addition, Cabrera-García et al. [124] reported the synthesis of engineered contrast agents in a single structure for T1–T2 dual magnetic resonance imaging using a thin and homogeneous silica coating through hydrolysis and polymerization of silicate at neutral pH. They produced $Gd(H_2O)_4[Fe(CN)_6]@SiO_2$ magnetic nanoparticles that are very stable in biological fluids. Thus, this product exhibits an important action as MRI CA, improving positive and negative contrasts in T1- and T2-weighted MR images, which can be applied for both in vitro and in vivo models. Moreover, this study showed that biosafety profile and excellent capacity to add in organic molecules, showing remarkable potential for medical diagnostic tools.

### Magnetic Hyperthermia Therapy and Magnetic Resonance Imaging

Magnetic nanoparticles are commonly studied for magnetic hyperthermia therapy (MHT) and as contrast agents for magnetic resonance imaging (MRI). The responsiveness of MNPs to an external magnetic field (MF) is useful in cancer treatment due to the ability to target the drug to the organs and tissues of interest, decreasing the non-specific toxicity of chemotherapeutic agents [126]. Iron oxide nanoparticles (IONPs) are a class of nanoparticles widely explored [127,128]. These nanoparticles are known to exhibit magnetic properties only in the presence of the external magnetic field. Moreover, IONPs can be easily synthesized and chemically modified [90]. However, considering the critical size and high surface energy, magnetic nanoparticles tend to aggregate [50]. Thus, surface modification with silica-based nanoparticles can not only increase stability but also become magnetic nanoparticles even more functional and biocompatible [129].

The advantage of the use of magnetic hyperthermia therapy (Figure 7) is the higher sensitivity of tumor cells to heat compared to healthy cells. The increase in local tumor tissue temperature (41–46 °C) causes cell damage for different pathways as well as sensibilize cells to drug chemotherapy [130]. Cell damage can occur by diverse pathways: i: through conversion of energy to heat (Neel relaxation); ii: mechanical damage caused by MNPs rotation in response to the external changing MF direction (Brownian relaxation) [45]; iii: production of heat shock proteins; iv: increase in immune response (recognition by cytotoxic T lymphocytes) [126].

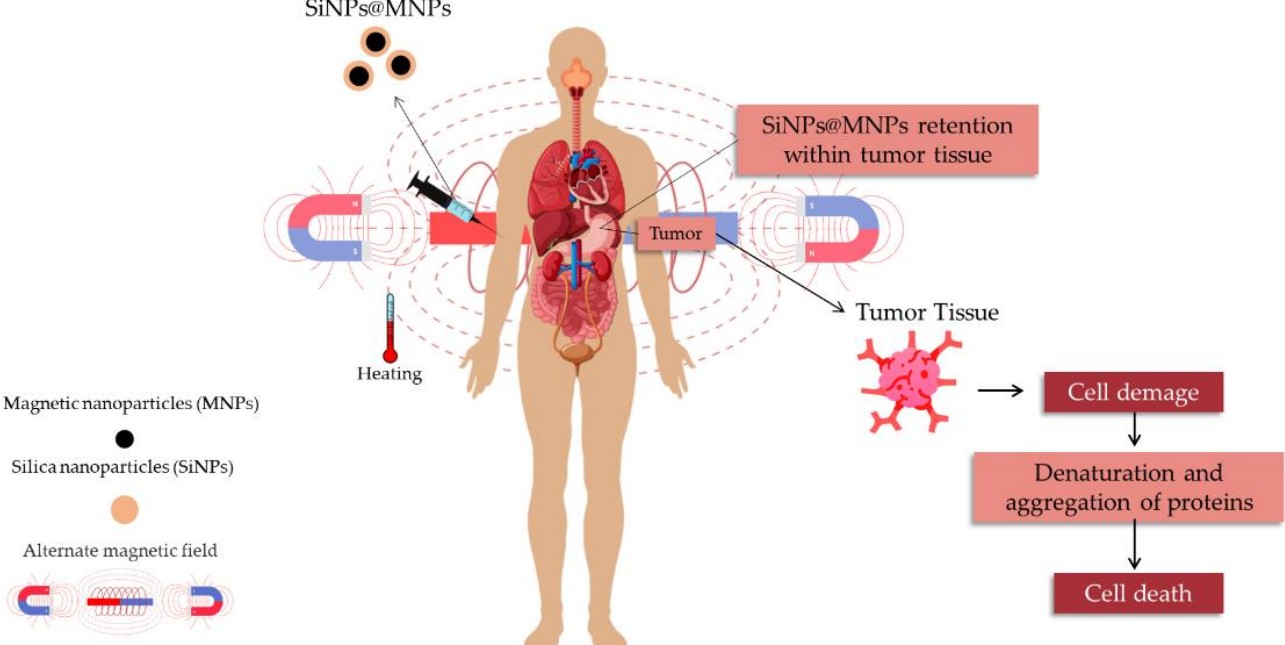

**Figure 7.** Magnetic nanoparticles as the therapeutic agents for MHT.

The MRI-based conventional diagnosis uses gadolinium derivatives as contrast agents. Gd is rare earth from the lanthanide series that shows high toxicity in its inorganic form. Therefore, this metal is employed in chelate forms to increase stability and consequently reduce its toxicity [131]. However, gadolinium-based contrast agents (GdBCAs) still trigger side effects such as gadolinium deposition disease, skin changes, neurotoxicity, nephrotoxicity, and pancreatitis [132]. Iron oxide nanoparticles have been widely explored in MRI, considering their superparamagnetic properties, high biocompatibility, and relaxivity [133].

Nanotheranostic is a novel concept involving nanoscience and nanotechnology that associates therapy and diagnostics. In the past, magnetic nanoparticles were employed for cell migrating detection through MRI, and the results obtained revealed an improvement in the resonance signal [134]. Similarly, Benyettou and coauthors [135] reported an interesting study regarding the use of organic framework-based magnetic nanoparticles for multitasking applications. According to the results, it was possible to verify a linear correlation between the nanoparticles concentration and contrast, improving the relaxivity values. The theranostic proposal allows targeting antitumor drugs using nanocarriers as well as verifying tissue accumulation, toxicity, and cell processes [135,136]. Aiming to overcome the limitations of chemotherapy, such as high toxicity and poor specificity, Wang et al. [45] developed an injectable self-healing hydrogel (Mn-Znferrite@mesoporous silica) for doxorubicin delivery in the tumor site. Doped ferrite with $Zn^{2+}$ and $Mn^{2+}$ can convert the $H_2O_2$ into ·OH radical (·OH radicals cause tumor cell death) through ion-mediate Fenton reaction. Moreover, the superparamagnetic properties of this self-healing hydrogel allowed its use in MHT and MRI. The results showed the excellent effect of magnetic hyperthermia and MRI. An increase in the cytotoxicity of doxorubicin against the

4T1 cell line was observed due to higher cell uptake triggered by the nanoencapsulation and association with an alternate magnetic field (AMF). The in vitro and in vivo MRI results demonstrated that the increase in iron concentration improves transverse relaxation. Moreover, magnetic nanoparticles can enhance image resolution due to their high relaxivity (303.9 mM$^{-1}$ s$^{-1}$). Magnetic silica-based nanoparticles with high relaxation values (175 mM$^{-1}$ s$^{-1}$) were also reported by Sheng et al. [95]. These results display the potential application of nanoparticles as a theranostic platform.

Effects on the relaxation time caused by iron concentration were reported by Kubíčková et al. [26]. The MRI experiment revealed that an increase in iron content caused a reduction in the signal, which can be explained by the high relaxivity. Furthermore, $MnFe_2O_4$ and $MnFe_2O_4@mSiO_2$ nanoparticles were also used as contrast agents (CAs) [25]. The results of the MRI showed that pristine manganese ferrite exhibits higher capacity than the CAs (transverse relaxation-T2) due to relaxivity value (1.04 μg mL$^{-1}$ s$^{-1}$). Regarding the MHT study, the results revealed that the coating with silica nanoparticles decreased the specific absorption rate (SAR). The heat dissipation mechanisms reported are the Neel and Brown relaxation due to the critical size of magnetic nanoparticles.

Effects of the alternate magnetic field were also evidenced by Pon-on et al. [49] when employing Dox-carried into MagSi@Chi-g-NIPAAm. The internal stimulation increased the Dox release. Under the same experimental conditions (pH 4.0), it was observed that the drug release rate improved by around 30% with AMF application.

Moorthy et al. [28] produced a mesoporous silica nanocarrier system (FeNP@SiOH@CET) modified by a crown ether triad unit (CET) focused on drug carriers, especially chemotherapy and heat therapy. This system presents a potential application for cancer therapy to reduce the side effects of pH-responsive drug delivery and magnetic hyperthermia. The surface-modified CET gating units protect the encapsulated drug inside the mesopores of the FeNP@SiOH@CET NPs system under a physiological environment and exhibit a pH-responsive release in acidic conditions promoted by an alternating magnetic field (AMF). This system presents action against MDA-MB-231 considering magnetic hyperthermic heating induced by the exposed AMF. Therefore, FeNP@SiOH@CET NPs present an important potential for cancer treatment combined with magnetic hyperthermia therapy applications.

Iliasov et al. [121] investigated the in vitro cytotoxicity of PEGylated MNPs and bioinert silica-coated MNPs on tumor cells (PC3-human prostate cancer) using a low-frequency alternating magnetic field (LF AMF). After 48 h of incubation, it was verified that the cytotoxic effects occurred in a dose-dependent manner and increased with coating. The amplitude of LF AMF and preincubation time had a significant contribution to the cytotoxicity, while the frequency did not affect cell proliferation. Similarly, Yang et al. [110] reported that the FDPOMs complex improved MR tumor imaging ability and showed excellent anticancer action considering the intense magnetic targeting capacity under an external magnetic field.

## 4. Conclusions

Silica is approved by the Food and Drug Administration for its outstanding characteristics, such as inertness and biocompatibility. Additionally, it is used to improve the physicochemical and thermal stability of other materials and drug delivery systems. In particular, silica is used in the coating of magnetic nanoparticles, protecting the magnetic core from leaching and erosion as well as decreasing the surface energy, which is related to agglomeration in the biological environment. Numerous studies demonstrated the remarkable biological activity of silica nanoparticles and silica-based nanoparticles. These reports include drug release and delivery studies aiming to decrease systemic toxicity and side effects, increasing the efficiency and stability of chemotherapeutics. Innovative treatments include involving pH-thermal responsiveness, alternate magnetic field, and combining therapy and diagnosis, especially in nanotheranostics, which has become an interesting field of research considering the broad spectrum of applications already demonstrated as potential and promisor advances in cancer therapy. Nevertheless, extensive progress in

studies for daily basis application is required. Notable aspects related to antimicrobial and antioxidant applications were reported in this review, such as the chemical modification, concentration, and specificity (bacteria cell constitution).

Studies regarding safety profile and biocompatibility show the future challenges related to nanoscale materials. Although nanoparticles have excellent characteristics to overcome the disadvantages of conventional materials and treatments, there is still much to explore about the behavior of nanoparticles in the biological environment and their long-term effects. Another important point refers to a more detailed investigation of the interaction with healthy cells evaluated through animal and alternative models (mechanisms of cellular uptake, excretion, and bioaccumulation. Moreover, from a perspective, the importance of knowledge and approach to the ecotoxicity of these nanoparticles is highlighted.

**Author Contributions:** F.d.S.B., writing—review; F.B.N., Conceptualization; T.d.R.S., C.F. and F.C.C., writing; C.R.B.R., conceptualization, writing—review editing, supervision. All authors have read and agreed to the published version of the manuscript.

**Funding:** This research received no external funding.

**Institutional Review Board Statement:** Not applicable.

**Informed Consent Statement:** Not applicable.

**Data Availability Statement:** Not applicable.

**Acknowledgments:** The authors thank Laboratório de Materiais Magnéticos Nanosestruturados— LaMMaN, CAPES, CNPq, FAPERGS, and support.

**Conflicts of Interest:** The authors declare no conflict of interest.

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
