# Peer review of "Biological Applications of Silica-Based Nanoparticles"

_magnetochemistry, doi:10.3390/magnetochemistry8100131_

Round 1

Reviewer 1 Report

26 confinment
35 biocompatibility - is it neutral feature? it is not clear function in the topic described, how can it be related to biomedical applications, it fits to sockets for example made from natural polymers
35 protecting - is to narrow meaning, I mean to what extend?
35 stability in acidic (acid perhaps) conditions - maybe should be in tissue
40 covalent bondings
47 aggregation of magnetic beads - is it always? Just not sure. In the case of metals it depends on polarity.
54 [18-24] magnetic nanoparticles - it is too wide statement. Some aaditional accompanying features should be mentioned.

Also very good pictures requires info about software applied or licence number if any

I am not sure if word adapted is sufficient statement to present other people results?

After such comments, I do not feel that this manuscript is suitable enough for medical applications help even in presented theoretical form. I am sorry.

Author Response

We thank you for the opportunity to submit our revised paper to the Magnetochemistry. We revised all the manuscript with great attention. Your comments provide us with detailed and very useful reports. The modifications are described above and the modifications in the final version of the manuscript. By addressing their comments in the revised version, we are confident that the paper has been considerably improved, thus shaping it to a publishable form. Please, find below the answers to the referee's comments.

With kind regards

Prof. Dr. Cristiano Rodrigo Bohn Rhoden

Queries are given in black and answers in blue.

Comments and Suggestions for Authors

26 confinment

A: corrected

35 biocompatibility - is it neutral feature? it is not clear function in the topic described, how can it be related to biomedical applications, it fits to sockets for example made from natural polymers

A: Biocompatibility refers to the relation between lower toxicity and physiological susceptibility. This property is widely and detailed discussed along the manuscript i.e., bioaccumulation, distribution, etc.

35 protecting - is to narrow meaning, I mean to what extend?

A: Thank you for the observation. The sentence was rewritten (See Lines 51-52).

35 stability in acidic (acid perhaps) conditions - maybe should be in tissue

A: Thank you. The sentence was corrected.

40 covalent bondings

A: Corrected. Thanks.

47 aggregation of magnetic beads - is it always? Just not sure. In the case of metals it depends on polarity.

A: Thanks for the correction. In fact the aggregation normally occurs due to the critical nanoparticles size, considering Van der Walls interaction, nevertheless not always. The sentence was corrected by adding “generally” (See Line 89).

54 [18-24] magnetic nanoparticles - it is too wide statement. Some aaditional accompanying features should be mentioned.

A: Done. Thank you. (See Lines 87-88)

Also very good pictures requires info about software applied or licence number if any

A: Thank you. In all figures the software and version were included.

I am not sure if word adapted is sufficient statement to present other people results?

A: Thank you for the observation. Considering the short time-window for correction and submission, we decide to maintain the citation although removing the figure.

After such comments, I do not feel that this manuscript is suitable enough for medical applications help even in presented theoretical form. I am sorry.

A: Thank you for the sincere observation. We understand your point of view, nevertheless, the main goal of the manuscript was to assemble recent biomedical studies/applications related to SiNPs, to support future medicinal advances, as well as, in some extend to stimulate futures studies to circumvent their limitations.

Reviewer 2 Report

Silica is approved by the Food and Drug Administration (FDA) due to its inertness, low toxicity, thermal resistant characteristic, and biocompatibility. The presence of silanol groups (Si-OH) in its structure allows easy complexation with other materials by covalent functionalization, such as antibody, nucleic acids, and fluorescent molecules. In biological systems, nanosilica and particles incorporated into it show certain specificities, i.e., selectivity. For example, the size of a silica nanoparticle can selectively enter some organs, relatively large particles from the bloodstream, due to the pores on the capillaries, can only enter the liver and spleen. Chemical modification of the OH groups of nanosilica particles achieves (modifies) the selectivity towards certain organs, or the stability of the particle - change in bioavailability of the incorporated drug.

This review article is comprehensive and very well designed regarding the application of silica nanoparticles.

Weakness of the manuscript:

In one chapter, the authors could give a brief overview of the chemical transformations of silica nanoparticles, as well as the instrumental methods used to characterize the size (nanoparticles), stability, etc. of these particles.

Author Response

We thank you for the opportunity to submit our revised paper to the Magnetochemistry. We revised all the manuscript with great attention. Your comments provide us with detailed and very useful reports. The modifications are described above and the modifications in the final version of the manuscript. By addressing their comments in the revised version, we are confident that the paper has been considerably improved, thus shaping it to a publishable form. Please, find below the answers to the referee's comments.

With kind regards

Prof. Dr. Cristiano Rodrigo Bohn Rhoden

Queries are given in black and answers in blue.

Comments and Suggestions for Authors

Silica is approved by the Food and Drug Administration (FDA) due to its inertness, low toxicity, thermal resistant characteristic, and biocompatibility. The presence of silanol groups (Si-OH) in its structure allows easy complexation with other materials by covalent functionalization, such as antibody, nucleic acids, and fluorescent molecules. In biological systems, nanosilica and particles incorporated into it show certain specificities, i.e., selectivity. For example, the size of a silica nanoparticle can selectively enter some organs, relatively large particles from the bloodstream, due to the pores on the capillaries, can only enter the liver and spleen. Chemical modification of the OH groups of nanosilica particles achieves (modifies) the selectivity towards certain organs, or the stability of the particle - change in bioavailability of the incorporated drug.

This review article is comprehensive and very well designed regarding the application of silica nanoparticles.                                                                              

Weakness of the manuscript:

In one chapter, the authors could give a brief overview of the chemical transformations of silica nanoparticles, as well as the instrumental methods used to characterize the size (nanoparticles), stability, etc, of these particles.

A: The topic was included. Thank you for the observation. It was really helpful to complement the manuscript. (See Lines 113-221).

Reviewer 3 Report

Current manuscript entitled Biological applications of silica-based nanoparticles”
by “Bruckmann et al” reviewed on the recent advances of silica-based magnetic nanoparticles focusing on drug delivery and drug target systems but also the use in magneto hyperthermia and magnetic resonance imaging. Further provided an overview of the challenges and perspectives related to use of silica-based magnetic nanoparticles in the biomedical field. Manuscript seems interesting and written well. The review article can be accepted after addressing the following comments.

1.     In the introduction authors staed that “Among the nanoparticulate systems, silica nanoparticles (SiNPs) stand out for their excellent properties for biological applications” Please explain them

2.     Some potential literature is dedicated on the nanomaterials for toxic pollutants and hazardous constituents that can provide a basic idea; those must be cited.

https://doi.org/10.1016/j.tifs.2021.11.018

3.     Please check thoroughly for the Grammatical errors

4.     Provide the challenges that are currently facing with the “Silica based nanoparticles”

5.     Does silica based nanoparticles have any role in the point of care detection. If so discuss about them.

6.     Remove (N&N) on line number 24

7.     In the introduction discuss about the related review articles and their drawbacks.

Author Response

We thank you for the opportunity to submit our revised paper to the Magnetochemistry. We revised all the manuscript with great attention. Your comments provide us with detailed and very useful reports. The modifications are described above and the modifications in the final version of the manuscript. By addressing their comments in the revised version, we are confident that the paper has been considerably improved, thus shaping it to a publishable form. Please, find below the answers to the referee's comments.

With kind regards

Prof. Dr. Cristiano Rodrigo Bohn Rhoden

Queries are given in black and answers in blue.

Comments and Suggestions for Authors

Current manuscript entitled Biological applications of silica-based nanoparticles”
by “Bruckmann et al” reviewed on the recent advances of silica-based magnetic nanoparticles focusing on drug delivery and drug target systems but also the use in magneto hyperthermia and magnetic resonance imaging. Further provided an overview of the challenges and perspectives related to use of silica-based magnetic nanoparticles in the biomedical field. Manuscript seems interesting and written well. The review article can be accepted after addressing the following comments.

1. In the introduction authors staed that “Among the nanoparticulate systems, silica nanoparticles (SiNPs) stand out for their excellent properties for biological applications” Please explain them

A: The properties of SINPs such easy functionalization (normally without need of unconventional reagents), lower toxicity (FDA use, approved), and high surface area, among others were better quoted and detailed along the manuscript.

2. Some potential literature is dedicated on the nanomaterials for toxic pollutants and hazardous constituents that can provide a basic idea; those must be cited.

https://doi.org/10.1016/j.tifs.2021.11.018

A: Reference was added (see Line 45).

3. Please check thoroughly for the Grammatical error

A: Thank you for the observation. The English grammar was checked in the entire manuscript.

4. Provide the challenges that are currently facing with the “Silica based nanoparticles”

A: Thank you for the suggestion. The topic was added (See Lines 60-86).

5. Does silica based nanoparticles have any role in the point of care detection. If so discuss about them.

A: Thank you for the observation. The topic Point of care was included (See lines 868-890).

6. Remove (N&N) on line number 24

A: It was removed.

7. In the introduction discuss about the related review articles and their drawbacks.

A: Thank you for the suggestion. The drawbacks topic was added (See lines 60-85)

Reviewer 4 Report

The publication tries to cover a very broad topic. Related research is almost impossible to cover. The authors tried to cover as many results as possible. However, some subsections would need to be expanded.

2.1 Biomolecule purification subsection can be expanded with more examples such as: https://doi.org/10.3390/molecules24224146, http://dx.doi.org/10.1016/j.jcis.2011.09.051, http://doi.wiley.com/10.1002/anie.201003602

Although many medical and biological applications are beyond the reviewer's field of expertise, feels that the important areas and results have been presented in the review publication.

Author Response

We thank you for the opportunity to submit our revised paper to the Magnetochemistry. We revised all the manuscript with great attention. Your comments provide us with detailed and very useful reports. The modifications are described above and the modifications in the final version of the manuscript. By addressing their comments in the revised version, we are confident that the paper has been considerably improved, thus shaping it to a publishable form. Please, find below the answers to the referee's comments.

With kind regards

Prof. Dr. Cristiano Rodrigo Bohn Rhoden

Queries are given in black and answers in blue.

The publication tries to cover a very broad topic. Related research is almost impossible to cover. The authors tried to cover as many results as possible. However, some subsections would need to be expanded.

2.1 Biomolecule purification subsection can be expanded with more examples such as: https://doi.org/10.3390/molecules24224146, http://dx.doi.org/10.1016/j.jcis.2011.09.051, http://doi.wiley.com/10.1002/anie.201003602

A: Thank you for the observation. The Suggested subsection with the articles were included (See Subsection 3.1 , Lines 223-246).

Although many medical and biological applications are beyond the reviewer's field of expertise, feels that the important areas and results have been presented in the review publication.

A: Thank you.

Round 2

Reviewer 1 Report

I have changed my mind in regards to acceptance or not for positive evaluation output

Name of softwere you use is sufficient for reviewer. If it is recommended hide the name. Just I know it. But it is up to you.

Specific remarks

line: 589 nanoparticles carriers would be better

lines: 642, 645 and 688 HeLa cells (what catalog number of cells or specific name if info was provided)

If you could extend the interestig term "teranostic" treatment and diagnostic. Is it in several variants notifird in papers published or more than several numbers up to date?

Also any critical review of such strategy would be of great importance for readers. is if future or just more specific or case-dependent achievemet(s) in regards to cell-line experimets in the lab or clinical use with patients.

Author Response

We thank you again for the opportunity to submit our manuscript. We are glad that you change your mind regarding the publication and also revised you last comments. The modifications are described above and the modifications in the final version of the manuscript. By addressing these final comments in the revised version, we are confident that the observations were attended. Please, find below the answers.

With kind regards

Prof. Dr. Cristiano Rodrigo Bohn Rhoden

Queries are given in black and answers in blue.

I have changed my mind in regards to acceptance or not for positive evaluation output

Name of softwere you use is sufficient for reviewer. If it is recommended hide the name. Just I know it. But it is up to you.

Specific remarks

line: 589 nanoparticles carriers would be better

A: Thank you for the suggestion. It was included.

lines: 642, 645 and 688 HeLa cells (what catalog number of cells or specific name if info was provided)

A: Unfortunately, we were no able to inform. Considering that the authors did not provide the catalog number, but the HeLa cells refer to the human immortal lineage of cervical carcinoma.

If you could extend the interestig term "teranostic" treatment and diagnostic. Is it in several variants notifird in papers published or more than several numbers up to date?

A: Thank you for the suggestion. We improve the theranostics on the main text, by adding some past/present data regarding thsi outstanding diagnosis alternative (See Lines 922-929).

Also any critical review of such strategy would be of great importance for readers.

A: A particular point of view, regarding theranostics application was added at the conclusions topic (See Lines 988-991).

is if future or just more specific or case-dependent achievemet(s) in regards to cell-line experimets in the lab or clinical use with patients.

A: The recent communications have been reported a promisor and potential applications of silica-based magnetic nanoparticles for drug delivery, magnetic hyperthermia, and magnetic resonance imaging, however a lack of studies exploring their effects on in vivo models are still remaining. As well-known in vitro studies are very important to determine the safety profile and biocompatibility. Nonetheless, they are not suitable for ensuring new therapy-based nanotechnology development considering the behavior of these nanoparticles in the biological environment is not well-established.

Thank you  very much for your interest and attention.
